# Lived experiences of recovered COVID-19 persons in Nigeria: A phenomenological study

**Friday E. Okonofua**[1,2,3]*, **Lorretta F. C. Ntoimo**[3,4], **Vivian I. Onoh**[1], **Akhere A. Omonkhua**[1,3,5], **Christiana A. Alex-Ojei**[4], **Joseph Balogun**[6]

1 Centre of Excellence in Reproductive Health Innovation (CERHI), University of Benin, Benin City, Nigeria, 2 Department of Obstetrics and Gynaecology, School of Medicine, University of Benin, Benin City, Nigeria, 3 Women's Health and Action Research Centre (WHARC), Benin City, Nigeria, 4 Department of Demography and Social Statistics, Faculty of Social Sciences, Federal University Oye-Ekiti, Oye, Nigeria, 5 Department of Medical Biochemistry, School of Basic Medical Sciences, University of Benin, Benin City, Nigeria, 6 College of Health Sciences, Chicago State University, Chicago, Illinois, United States of America

* friday.okonofua@cerhi.uniben.edu, feokonofua@yahoo.co.uk

## Abstract

### Background

Numerous publications have documented the mode of transmission and prevention of COVID-19 but little or no evidence exists on the experiences of people who survived the infection.

### Objective

This study explored the specific experiences of persons who were infected with COVID-19, but have recovered completely. A secondary objective was to identify essential elements in the lived experiences of such persons, which would be useful in designing appropriate policies and programs for managing the virus in Nigeria.

### Method

The data were collected using in-depth interviews with 21 persons who were diagnosed with the virus and recovered. The data were transcribed and analyzed qualitatively using NVivo software. The experiences of the survivors of COVID-19 were examined under six themes: compliance with prevention measures before being infected, perceptions on how they contracted the virus, the symptoms they experienced, the management of the disease, their experiences with the healthcare system, their emotional experiences, and their recommendations on specific strategies to prevent and manage the virus based on their experiences.

### Results

The commonly perceived means of contracting the virus were through colleagues, patients, and friends who were infected. The most commonly experienced symptoms were anosmia and fever. The health providers were described as courteous but some of the respondents observed avoidance and fear. Not all the interviewees knew the drugs they were treated with, but some, particularly the medical personnel, identified hydroxychloroquine, azithromycin, vitamin C, Augmentin, among others. Some of the participants used herbal remedies.

**Data Availability Statement:** All relevant data are within the paper and its Supporting Information files.

**Funding:** FEO received a grant from the African Centre of Excellence in Reproductive Health Innovation, grant # Team B/003/2020. The funders had no role in study design, data collection and analysis, decision to publish, or preparation of the manuscript.

**Competing interests:** The authors have declared that no competing interests exist.

While some respondents recounted good experiences in the isolation centre, others had unpleasant experiences. Direct and indirect encounters which were perceived as stigmatizing and discriminatory were reported by some respondents.

## Conclusion

We conclude that persons who recovered from COVID-19 in Nigeria had varied experiences relating to the mode of infection, the clinical features, methods of treatment, and psychosocial effects of the virus. These experiences would be useful for designing and implementing appropriate interventions, policies, and programs for managing the pandemic in the country.

## Introduction

The novel coronavirus disease 2019 (COVID-19) pandemic is a major public health concern globally. Since the virus was first discovered in December 2019 in Wuhan, China, it has spread to nearly all countries around the world [1]. As of March 3, 2021, over 144 million cases had been reported worldwide with 3.07 million deaths representing a case-fatality rate of 2.12% [1]. COVID-19 is caused by the novel severe acute respiratory syndrome coronavirus 2 (SARS-CoV-2) [2], and its spread has greatly exceeded similar outbreaks of coronaviruses such as the severe acute respiratory syndrome coronavirus (SARS-CoV) [3, 4] and the Middle East respiratory syndrome coronavirus (MERS-CoV) [5] in 2003 and 2012, respectively. Based on the known mode of transmission of COVID-19, control strategies proposed by the World Health Organization (WHO) have included mask-wearing, consistent handwashing, social distancing, and lockdowns to reduce the chances of people passing on the virus from one to another [6]. Available evidence shows that these preventive measures are effective [7]. However, no effective drugs have yet been identified, while several vaccines were recently discovered and are currently in use [8].

A modelling study conducted in the USA suggests that if masks are universally worn in New York State, this could reduce the number of projected deaths over two months by 17–45% [9]. Other evidence shows that suboptimal or lack of handwashing elevates the risk of COVID-19, but proper hand hygiene could remove 97%-100% of the virus in the palm [10, 11]. Since the onset of the disease, Nigeria has established a Presidential Task Force, which has adopted diverse preventive measures including isolation of persons suspected to have contacted the virus and active contact tracing to limit community transmission. Despite these efforts, the number of cases continues to increase. According to the Nigerian Centre for Disease Control (NCDC), as of April 24, 2021, the number of confirmed COVID-19 cases in Nigeria has risen to 164,633, out of which 154,643 have recovered, while 2,061 have died [12]. With this number, Nigeria now has the fourth highest number of confirmed cases of coronavirus infection in Africa.

Numerous publications have documented the mode of transmission and prevention of COVID-19 [13–17], but no evidence exists in Nigeria on the experience of people who survived the infection. Our study documented the lived experiences of persons who recovered from COVID-19 infection in Edo State, Nigeria. Documenting the experiences of recovered persons will contribute to providing information on the social context under which the virus was contracted, the nature of the symptomatology and clinical features of the disease, the methods used for treatment, and the perceptions of affected persons on the responsiveness of the health care system in tackling the virus. We believe the results of the study will be useful for

strengthening health systems interventions and policies for curtailing the virus in Nigeria and other low resource settings.

## Methodology

### Study design

This study adopted a phenomenological approach as described by Groenewald Thomas [18] to explore the lived experiences of patients who recovered from COVID-19 infection. This approach was adopted considering the personal and sensitive nature of COVID-19 infection and the need to elicit in-depth information on the experiences of the participants. The data were collected using unstructured in-depth interviews with persons who recovered from COVID-19 infection. The consolidated criteria for reporting qualitative research (COREQ) were used in the presentation of the manuscript (See S1 File).

### Participants and recruitment

The inclusion criteria were adults age 18 and above, diagnosed of COVID-19 infection with PCR laboratory testing, treated in one of the government-designated treatment centres, and had tested negative for the virus and certified recovered by care givers. We first recruited the first cohort of recovered persons from the University of Benin Teaching Hospital, one of the certified COVID-19 testing and isolation centres approved by the Nigerian Centre for Disease Control (NCDC). There-after, we used the snowball sampling technique [19], with the initial participants recommending and contacting those they know that similarly recovered from the virus. The objectives and methods of the study were fully explained to the participants through telephones, emails, or WhatsApp and their willingness to participate were solicited. They were told that the results would be used only to elicit information on ways to better handle future cases of the pandemic, that all information would be held in confidence, and that they were free to refuse to answer any of the questions.

Twenty-one interviews were successfully achieved–four persons refused to participate due to time constraint and unwillingness to share their experiences. The sample size of 21 was largely guided when evidence of data saturation was obtained. The profile of the participants is shown in Table 1 below.

### Data collection procedures and setting

All the interviews were conducted by the lead author (FEO) who is a Professor of medicine with over 30 years of experience in health research, and data collection assistants (male and female) who had obtained at least a first degree in Social and Health-related Sciences. They were trained in the art of collecting qualitative data. All the interviews held in venues conve-nient for the interviewees, and through Zoom/telephone between July and October 2020. Each interview was completed during the first appointment, and the average duration of the inter-views was 40 minutes and each interview ended when no further issues arose. All interviews were conducted in English, and audio-recorded with the permission of the interviewees. All the precautionary guidelines for the prevention of COVID-19 infection stipulated by the NCDC were observed during the face-to-face interviews.

Drawing from the illustration of phenomenological design by Groenewald (2004) [18], an interview guide was drafted by FEO for data collection with input from LFCN and JB who are experienced in qualitative research (See S2 File). It consisted of an icebreaker question, fol-lowed by the perceived mode through which the virus was contracted, experience with testing, discovery, isolation and treatment, social and financial effects, perceptions about management and treatment of coronavirus, and recommendations.

**Table 1. Profile of the participants.**

| Archival number | Age | Sex | Profession |
| --- | --- | --- | --- |
| IDI001 | 56 | Male | Lawyer |
| IDI002 | 35 | Female | Medical Doctor |
| IDI003 | 28 | Male | Medical Doctor |
| IDI004 | Not provided | Male | Medical Doctor |
| IDI005 | 44 | Male | Lecturer |
| IDI006 | 33 | Male | Medical Doctor |
| IDI007 | 30 | Male | Nurse |
| IDI008 | 30 | Male | Medical Doctor |
| IDI009 | 26 | Female | Medical Doctor |
| IDI010 | 32 | Male | Medical Doctor |
| IDI011 | 49 | Male | Medical Doctor |
| IDI012 | 49 | Male | Banker |
| IDI013 | Not provided | Male | Not provided |
| IDI014 | 56 | Male | Oil and Gas worker |
| IDI015 | 34 | Male | Lawyer |
| IDI016 | 31 | Male | Paramedic |
| IDI017 | 41 | Male | Medical Doctor |
| IDI018 | 31 | Male | Paramedic |
| IDI019 | 36 | Female | Lawyer |
| IDI020 | 40 | Male | Police Officer |
| IDI021 | 28 | Female | Housewife |

## Data analysis

All the interviews were transcribed verbatim, and validated using member check. Coding was conducted by CAA using NVIVO 12, computer-aided qualitative data analysis software. A second data analyst (LFCN), and the Principal Investigator (FEO) validated the coding. The data analysis followed the deductive and inductive approaches to thematic coding. Codes were generated from the interview guides and the project objectives, and the themes emerging from the narratives. Each transcript was coded after reading the transcripts several times to become familiar with the data. Similar codes were merged, and all codes were grouped into sub-categories and themes.

The results of the analysis are presented narratively in 14 themes with apt quotations showing the participant's unique identifier. The main themes include precautionary behaviour before COVID-19 infection; perceived means of contracting the virus; symptoms experienced; pre-existing health conditions; management and treatment by health providers, presented using three sub-themes on the attitude of the health providers to patients, testing, results and response to the results; and treatment regimen. Other main themes are perceived effectiveness of medical care; post-infection treatment regimen; supplementary and herbal therapy; experience in the isolation Centre; emotional effect/response; family involvement and response the experience of stigma and discrimination; financial cost; and recommendations.

## Ethical issues

The permission to conduct this study was obtained from the Ethical Review Committee of the University of Benin College of Medical Sciences, Benin City, Nigeria (CMS/REC/2020/286). There was no prior relationship between any of the researchers and data collectors and the participant before the study. All the participants were duly informed of the purpose of the

research, potentials risk and benefits. They were all assured of the confidentiality of the information they provided. Written informed consent was obtained from all the participants at the time of recruitment. Only those who consented were interviewed. In compliance with qualitative research ethics, direct identifiers were removed from the analysis.

## Results

### Precautionary behaviour before COVID-19 infection

Although many of the interviewees observed the preventive measures, such as wearing a mask, gloves, regular handwashing with soap, and maintaining physical distance, before they were infected, some of them could not say with certainty that their adherence was total. There were lapses such as when discussing in the office, at clubs, and in emergencies especially for participants who are medical personnel. Some of the interviewees who travelled abroad observed self-isolation on return to the country. A few (n = 3) did not observe any of the preventive measures before being infected.

> I was using face masks regularly and most times when we even have to go into the holding ward where we have highly suspicious patients, we even double our masks. Sometimes, I use face goggles, latex gloves, regular hand wash as often as possible . . . because we handle a lot of patients daily and sometimes some things just happen so fast. Sometimes patients come into Accident & Emergency, they are unconscious, some of them with features that we can. . .., before you later realize that this patient could be positive; we have done a lot. And sometimes you just wish that you could have done a lot to protect (IDI003).

> I was always doing all those things; the only thing I could say, maybe I broke protocol you understand in the clubhouse when you want to drink beer you will remove the mask you understand that was where I am m sure I missed it (IDI010)

> I also told my aides they had to get prepared to see me that nobody should come to see me and that they had to get their sanitizers. My friends' aides that came to pick me at the airport apparently were packaged for resistance. They had their sanitizers, facemasks so we didn't have contact and they didn't touch any of my things. My luggage as they were coming out, weresanitized. From the time I left them for the luggage department at the airport, I never touched them until I was isolated (IDI001).

> Yes, I did all the washing of hands, use of sanitizers and wearing of face masks . . . To the best of my knowledge, I think the infection is airborne because we put on our facemask all the time and remove it when we want to eat, probably it is through that process I got it (IDI007).

### Perceived means of contracting the virus

Most of the respondents perceived that they contracted the virus through contact with colleagues, patients, and friends who were infected. Other perceived means of contracting the virus included political meetings, attendance at club houses, droplets from infected persons inside public transport, customers in the bank, handshakes, in the office, and during foreign travels. A few (n = 2) of the respondents perceived that they contracted the virus when they travelled abroad. Only three had recent local travel history before being infected. Some others could not explain how they contracted the virus.

> Yes, I actually contacted it in the UK (IDI 001)

*No, I didn't travel outside the state, I was on lockdown before that incident. A woman sneezed in the bus and that was how the droplets fell on me and I thought I cleaned it off, I didn't know that I picked it up because I was not wearing a face mask. When I got home, I took my bath and all that, then a week after that I started having symptoms of it (IDI 021).*

*So, we had a patient and I was asked to give the patient some sample bottles for investigation. So he was flagged for COVID-19, but he was supposed to be in the holding area, but he was in the charade area and he wasn't on a face mask though his wife had one on. So, when I got there the patient was coughing. I then told his wife please ma can you put on the face mask for your husband?She explained that her husband could not cough very well with the face mask on. That was the last patient I came in contact with before I started having symptoms (IDI 002).*

*I had contact with a patient who was positive, and I also had contact with a couple of doctors who were positive, I cannot really pinpoint who I contracted it from. At the time of contact, they were not aware that they were positive because we tried our best to maintain the normal precautionary measures—wear a face mask, using hand sanitizers, and hand gloves. Again, I had about 3 patients who turned out to be positive and 2 doctors' colleagues who became. So, I could have gotten it from either of them (IDI 006).*

*The issue of how I contracted the virus is still a mystery. I had a travel history. I went to Germany and while in Germany, I checked the weather when I was going, I felt the temperature was moderate, not so cold. Our partners in Frankfurt gave a light weather forecast for which I took two cardigans but when we got there, my colleagues went out and started buying thick clothes that the cold was beyond what they had planned, and we stayed there all through. We had some tourist exposure; we went to some sites in Germany, and we even went to about five cities in Germany, including Luxemburg, Berlin, Frankfurt and one other place (IDI 005)*

*Well, I work in the bank and I attend to customers. So, I'm very sure, I don't know who I may-have contracted it from. I don't know, I don't know (IDI 012)*

*I don't know because right from when the regulations were put in place, I have been observing them—the face mask, the washing of hand and social distance. So, I was surprised when I was diagnosed with the virus. But the best I can think of is maybe money, buying things, market, bank and all (IDI 018)*

## Symptoms experienced

Although a few respondents were asymptomatic, most of them experienced at least one of the symptoms of the coronavirus infection. The symptoms reported by the respondents included anosmia, bodily weakness, fever, anaemia, breathing difficulty, cough (dry cough) and catarrh, diarrhoea, chest pain, rigor, headache, sore throat, sneezing, malaria-like symptoms, cold symptoms, pains, and pneumonia. Anosmia and fever were the most common symptoms experienced by the participants.

*Interviewer: What type of symptoms did you have*?

*Cough, catarrh which I thought was normal catarrh. Three days after that incident I started having catarrh, then later my throat was dry, initially, I was having scratchy throat and I was administering lemon and all those things because we had them at home. After a week it became serious with a headache and I was having a fever that refused to go. So, I had to go home and went to do a test around Irokpota hall because the day I got to my house I had to*

*treat for malaria, whenever I take malaria drug it works immediately but I finished the whole drug, but this time it did not work. My younger brother then suggested that I should go for COVID test which was why I went there. I was not able to feel any smell and that was what convinced me that it could COVID, the loss of smell and taste. So, when I got there, they told me that I should wait for the result which they later confirmed to be COVID (IDI 021)*

### Pre-existing health condition

Three of the interviewees had existing health conditions such as pneumonia, and diabetes.

*I could feel that cold entered my lungs because I have had a history of pneumonia in 1994 while fishing with my uncle. So, it was a similar experience I had when I felt that day. So, when I got back to the hotel, I put on the heater, took a lot of water. . .I have learnt to manage it even in my university days once I have cold symptoms I normally go for treatment and take some medications (IDI005).*

### Management and treatment by health providers

The narratives under this theme are presented in sub-themes–the attitude of health providers to patients; testing, results, and response to the result; and treatment regimen.

### Attitude of the health providers to patients

The providers were described as courteous but some of the respondents observed avoidance and fear. A few described the attitude of the providers to patients as excellent.

*The only thing that disturbed me of course was the attitude of the medical personnel—the nurses and doctors were all afraid. The reaction of the doctors was quite disturbing. At that time I will say I was rated the 2nd or 3rd case in Edo State, after the speaker announced his own so the whole world. The reaction and tension in the hospital was aggravating as the whole department had gone down on isolation with all contacts being tracedIt was the embarrassment that people were suspicious of everyone that was most disturbing (IDI 005).*

*Yes, it was very excellent I was happy with the way and manner they attended to me. I thought it was because I am a staff of the hospital but then my next-doorneighbourwho wasn't staff, received the same treatment that confirmed to me that the staff were doing very well. They will always reassure you, give you your drugs, and give you whatever food you wanted. The professionalism was top notch (IDI 019).*

### Testing, results, and response to the result

There were reports of confusion of the symptoms with the symptoms of pneumonia and confusion over the test results. In some cases, the test result was never seen; it was just announced to the patient. In other cases, it was inconsistent or came after the patient had exited the treatment programme. The handling of the test resulted in doubts about the veracity of the result by some of the study participants.

*. . ..so, I was taken to the isolation centre it was there they did the second test and said it was positive. I didn't even see anyone of the results that is why I am even skeptical there was no*

*result they only just announced to me which is a funny practice. At a point the testing became a laughable thing today positive on the 3rd day or so it will become negative and they have been receiving treatment and all that so one could doubt the validity of the test. In all thse, nobody was given any test results (IDI 005)*

*Interviewer: What was your reaction when the person told you your status?*

*Respondent: I just laughed*

*Interviewer: you just laughed why?*

*Respondent: because they were calling 16 days after I tested . . .*

*Interviewer: who then treated you? Did you isolate your self during the period after testing?*

*Respondent: I did not because I was not aware, it was after they told me I was positive that I tried to do some isolation but then it was already past 16 days, but I was not aware that I was COVID-19 positive for the first 16 days because the result didn't come out on time (IDI 010)*

*. . . because there was something that was really–I really didn't know what was happening. The test came positive on Monday, I moved to the isolation centre on Tuesday morning because I was told to come so that test would be repeated on Tuesday. So, when the test was repeated, the result came out on Thursday, it was negative. So, they took another sample on Friday and that one came out on Sunday and was positive. It was having this negative, positive, negative, positive results about 4 times before I had two negative results. I was very worried about why the test result was changing from negative to positive (IDI 017).*

Response to the positive results was mixed. To some, the positive result was devastating; it brought confusion, distress, shock, and fear. A few did not believe the result. Others who expected the positive result received the news more lightly whereas some just had a strong conviction that they would recover since they had no underlying ailment, were not old, and knew that the fatality rate of COVID-19 is not high. Others stated that they were not afraid because, by the time the test result came, they had started recovering. One interviewee responded with anger. Asked why, he explained that he suspected the infection but was asked to take a second dose of anti-malarial treatment. His anger was because they delayed the diagnosis and he had encountered many people between the time he was exposed and the time the diagnosis was made.

*Interviewer: So, how did you feel when you knew you were positive?*

*Respondent: Yes, well it was more of the psychology, in fact, I lost balance. The fact that the usual approach my wife was giving to it, when she came around, I was like no, no, no and she was like what is it and I told her that they called me, and it is positive. So, we were never the same again, we were thrown off balance. I was very worried, very, very worried. There was a time I went into the bath to brush my teeth and I didn't remember I didn't wash up and I just came out. So, I was feeling all sort of discomfort. So, I went back, and I noticed there was paste all over me, so you can imagine it was as bad as that. I just lost my senses, and it was very devastating (IDI001).*

*I broke down and I cried. I was eating and I stopped eating. So, I called my family and friends, so they came (IDI018).*

*At first, I would say the result coming out positive wasn't really a shock to me because the patient result was already positive, and I was already suspecting that I was positive. Where I*

*got so concerned was when I started having cough and chest pain. So, I felt very bad. I was down because I was thinking about what could happen. Am I going to die? Or am I going to have any organ damage? Or any of the severity I have been watching on TV. I had to read a lot of things online, basically. It was psychologically draining, and it was very terrible for the whole time even while I was in isolation I was down (IDI008).*

*When they called me that I tested positive, because first when we heard that one of our team members tested positive, I was afraid initially because I was close to the person we use to talk as in sit close to each other to discuss so I was afraid. When my result came out, the fear wasn't there because I expected it (IDI007).*

*I was really very sick then. You know, when you are very sick, you are too weak to even react. I just accepted it because the doctor came to me and explained to me that COVID-19 is not a death sentence that a lot of people have gone to the ward and alright now, they are fine and that I should just be patient that I will get well, that they are going to admit me in an isolation ward. So, I accepted it. I have always known that COVID-19 existed, so I accepted it (IDI015).*

Despite testing positive, one respondent (IDI020) did not believe he was infected. He suspected a foul play and believed it was all a ploy to satisfy a condition given to the doctors for payment.

*Interviewer: How did you think you got the virus?*

*Respondent: To be sincere I didn't even believe I caught the virus.*

*I thought there was foul play somewhere. The people made me to suffer. I spent 13 good days in isolation, so I just leave everything for God. I am back home and alive.*

*Interviewer: How did you feel tell me your general experience apart from being angry*

*Respondent: . . . I felt bad; they locked me somewhere for 13 days I did not have access to my children and wife. See, let me tell you the truth, the doctor know that nothing was wrong with me but the order they gave them was that they must get clients so that they can be paid. I am one of their clients because I know they used me to make money. In fact, among the people inside (because we were 10 inside the ward), I did not see anybody cough throughout the night. I did not see anybody who said catarrh was worrying them. They just tricked people so they can get clients. I believe thethave succeeded.*

## Treatment regimen

Not all the interviewees knew the drugs they were treated with, but some, particularly the medical personnel identified the following: hydroxychloroquine, azithromycin, vitamin C, intravenous antibacterial injection, chloroquine, Augmentin, erythromycin, dexamethasone, vitamin A, multi-vitamin, zinc, selenium, vitamin E, vitamin D, paracetamol, cough medicine, and floramo for diarrhea. For those who had other health conditions, they added the drugs for those conditions. For instance, magnesium phosphate was added for an interviewee who had an ulcer.

Asked what other thing could have helped them to recover, the common response was enough sleep, rest, drinking a lot of water, eating well, herbal drink, drinking warm water with lime, bitter leaf, among others.

*I think I started with Erythromycin, Hydroxychloroquine and Zinc Sulphate tablet. Then, Vitamin C. Actually, I started those ones at home before I moved to the isolation centre. It was just extended for like two weeks (IDI017).*

*I was treated with one injection something antibacterial they gave through intravenous and they also gave me Azithromycin and they gave me chloroquine that is what they gave me from the beginning of the treatment, they were doing every two days they were doing a blood test, they will test my kidney function, they were quite good, every 3hrs they will take my oxygen level and check my BP everything was stable they were commending my for my recovery. My recovery for them was one of the fastest and they shared it globally (IDI005).*

*Obviously, they won't tell you but with my experience, I would know because some of the doctors know me. So, one anti-malarial, and of course some zinc tablet, vitamin c, netilmicin, those were basically the drugs. After 4 days, I started getting better. The occurrence of fever started improving. The fever stopped. They came and took some other samples (IDI013).*

*. . . then, throughout the period, I never drank ordinary water, I had lime. At every point in time, I always took lime water. Then also the agbo [herbal drink for malaria] drink with dogoyaro(neem) leaf, guava, green tea and so many things. I also steam myself with it, breath in the heat and also drink it a lot. As hot as it is, I also took warm water all the time (IDI001).*

## Perceived effectiveness of medical care

Responding to the question on the effectiveness of the treatment, some of the respondents judged the treatment effective. They believed the management was effective because none of their contacts were infected, checks on patients were regular, and the treatment was free of charge. Others pointed out that the medication was effective because they recovered.

*Interviewer: will you say the management for the treatment of coronavirus especially in your case was effective?*

*Respondents: Of course, since they gave me drugs for six days and fed us for about 11 days, they tried. The government tried. Very much. We did not pay 1 kobo.*

*Yes, they were effective in their job they were coming every 3hrs to check my vital signs even though some new nurses that came later were doing it hurriedly, in my case I had my oxygen test kit that thing they used to read oxygen level, I had my own BP apparatus, I had my thermometer everything so if they come into my room I will place it for them and give them the readings some of them will avoid touching me. But I think they were quite up and doing I don't know how it is now but at that time there was so much fear among them. Where they need to do some work, for me, is when you go for a test and confirm this person is positive if not for anything they should put it in the file (IDI 005).*

A few did not consider their treatment effective. One of the respondents described an unpleasant experience of how he was picked from his house, and how he was transferred to another treatment centre like a "common criminal" in a bad ambulance, which he thinks, is used for conveying the dead because there was no seat. They reported that the providers were not there at night, and the patients were not given attention except the person became very weak.

*Well, something happened. I don't really like exposing this part of this process but in the midst of this restlessness and worry, they are not there at night . . . Now, first of all, I was told the doctors that came were so masked at the front of the house and that drew the attention of a lot of persons and they disappeared. The second group that came, came with an ambulance, I didn't know the status of the ambulance. They conveyed me in the ambulance to . . . I didn't like the experience. I felt I was conveyed like a common criminal. They dumped me at the*

*back of the ambulance with no chair for me to sit down. I think that ambulance was designed for dead people, so I just managed and the bad road . . . we spent nothing less than 3 hours. At a point, the ambulance stopped on the road, they didn't even come down to check me who was at the back of the ambulance. The purpose of the exercise was the person at the back, which was me, but I didn't know what they were doing, you know until later they moved on. It was a terrible experience.*

*. . . When I came out, I just told them that definitely, I wouldn't go back in this ambulance except I am dead and so would not care. Nobody was even saying sorry and I kept asking if that was how they were. . . I was feeling disappointed. The second point was that they brought me down and were leading me to a room. In fact, that was the height of embarrassment. The bed was so small with dirty bedsheets and pillow. The room was like a cubicle and can't be called a room, it was more like a store with a small window. The door at the back had no key and the outside was forest and bush. The toilet was not functioning, with no water and no light. They had to go bring water for the toilet. There was no health machine to monitor your body mechanisms, nothing was there. . . . (IDI 001).*

*Respondent: It was not good Sir.*

*Interviewer: Can you explain further? What do you mean by that?*

*Respondent: Ok when I came to the isolation centre the only time the people taking care of us came was the time, they came to collect the sample. . . . we were practically abandoned in the isolation centre? We came for four good days none of the doctors came to check on us. . . . and even when we call most times, we don't get responses. . . . Nobody brought drugs for us when we came in, we had to call that we don't have drugs before they brought drugs. . . . (IDI 002)*

*Interviewer: What will you say concerning the management and treatment of coronavirus in your case?*

*Respondent: It is fake. Now, all of us know that it is fake.*

*Interviewer: Why did you say so?*

*Respondent: Government is not serious all the people working are not serious*

*Interviewer: When you say they are not serious what do you mean?*

*Respondent: Aha! why will you be calling somebody after 16 days that he has coronavirus which is not so. They are not serious. Ideally, if you do a test is it not supposed to come out immediately. You even asked me whether I saw my result, am I not supposed to be presented a copy of my result? Don't just mind them (IDI 010)*

## Post-infection treatment regimen

The study participants commonly reported post-discharge treatment regimen. The common regimen included multivitamins, cough syrup, garlic and lemon in hot water, lime and hot water. Other respondents observed none.

*Respondent: Well, to be frank, that day we were released, we were not given a pin. We were not given a prescription, so nothing.*

*Interviewer: So, you have not taken any other thing*

*Respondent: Not at all. I have not. Besides that I told you in the morning, I add garlic to hot water (IDI004)*

## Supplementary and herbal therapy

Asked if they supplemented the treatment with other forms of therapy, some of the participants reported increasing their intake of vegetables and fruits.

*Yea. Initially, when I suspected, I am not this fruit guy, I don't like fruits, but I had to. I had to start taking a lot of fruits (IDI 17).*

Some of the participants reported a belief in the efficacy of herbal therapy. Some of the herbs used include garlic and ginger, garlic and lemon, neem leaves and ginger, neem leaves and spices. Those who did not use herbs were uncertain about the constituents of the herbs.

*Interviewer: Did you find that effective*?

*RES: To me when I take it very early in the morning my body is beautiful and it's good that I am telling you because you must feel it. That garlic, lemon when I take it (IDI 004)*

*The only thing I remember I had an electric kettle in my room and there was this garlic no is it garlic or ginger that my wife sliced because it has become a case of what we don't know we just expand the scope what I do is just drop it inside hot water and drink they were not my medication but I was just taking them and occasionally I ate bitter kola later the doctor advised I should not be taking it (IDI 005).*

Another respondent gave a confident and lucid narration of the effectiveness of a local herbal concoction called "agbo" in Yoruba.

*I want to tell you something now. In fact, I told the professor who was to interview me but because of the elections and so I had no time. I will still go back to it. I will say it on camera. When I left the hospital, I was fine but not completely. Not 100% fit but I think I was about 70% -75% fit. I will give you my experience and that of my friend. Probably because I'm a doctor, I have never believed in herbal medicines. The medicine they call "Agbo" is effective. Go and write it down o. A friend of mine who came to see me had coronavirus, but he refused to go to the hospital. In fact, he was going to some small hospitals and they were giving him malaria treatment. I said go ahead with your "Agbo". He went ahead with "Agbo", in 4 days he started jumping. A colleague of mine, one woman and two other men. Even the man that was admitted into the hospital before me that was discharged just before I was discharged. He went back on this 'Agbo" and it did wonders. So, I can tell you that "Agbo" is effective. It is an effective treatment to the management of COVID-19. And why you are not seeing so many cases in Nigeria hospitals, I can deduce it to the use of "Agbo" because most of our local people are what they can afford. It's not expensive, it is cheap and you use it at a low cost . . . And since I survived coronavirus, I have been telling the world that corona is real. That's the truth. "Agbo" is an effective treatment for the management of COVID-19. I can say so because I have evidence. My woman was also infected, and she was not admitted to any hospital, "Agbo" did the magic. That is the truth,because when we did the test it was positive, and because she was using "Agbo", she wasn't as critical as my own. Although we bought some of the same drugs for her which she took. By the time I went for the final test. I was negative and she was negative. And she was never admitted to the hospital (IDI013).*

## Experience in the isolation centre

Many of the respondents were taken to the designated isolation centres in the State, usually from the hospital ward but a few self-isolated in their residences. Some of them recounted

good experiences in the isolation centre. They reported that the centre was clean, the bed was good and neat, all the precautionary measures were observed, and the providers, particularly the doctors were friendly. They interacted with other patients in the centre but from the recommended distance. However, many others had an unpleasant experience such as inadequate providers, lack of sufficient attention, poor bed condition, among others.

> *Interviewer: So, is there anything you want to tell me about your experience there*?

> *Respondent: Well, when I got there, I called my wife in the night. I said this place is clean. The place is clean, I just have to confess the isolation center is clean. Then the doctors, the people there, the nurses, the cleaners, everybody there are all friendly. Especially the two doctors. I can't even remember their names. There is somebody that has dialysis or something like that, like a lungs issue, I don't know. They will come in the morning, with the covid, I use to see them the way they use to feed this old man. I was wondering, this one "no be your Papa" [is not your father], look at the way you're handling this old man. They will feed and clean. In fact, they tried (IDI 012)*

> *I really don't know if we were coming from space because they were only coming in 3 times in a whole day, so when anyone is in distress, they just look through the glass; we are on our own. A lady needed a bed pan I had to help to give her the bedpan, a man fell from his bed, it was another patient that helped to bring him up. A man fell in the bathroom and he slept there. A woman whatever. . . this drip something. . .I had to put it back and I told you I don't know anything about medicine. A man by me was being given blood by 3am I have to be up to make sure I turn off that whatever thing. So, their excuse is that they don't have enough PPE so they can't just come in often, they can only come in at particular times, so that's why I started also helping to give food from the window. So, they will open the window and pass me the food and I will go and deliver the food. So, generally, they would have helped more because like that man that fell, he died eventually so I guess it would have been prevented, then someone that died when I had left, she used to complain about fan and cold, so there was a time we had to turn off the fan for her, boil water. you know this heater. this bag. So, I told you I was the only one that was able to move around, so when I left, I got a call from a man that was there, he told me that the woman I was helping died, I said why? Was it that there unable to give her the hot whatever and turn off the fan . . .? So, I met her there, that means she had been there for a while and I left her there, and I believe she would have overcome this thing if she had more help, she coughed all through, they gave her many bottles of cough medicines. . . . they said they don't have enough PPE, and . . . the bathrooms were not very well maintained. There were people who broke down mentally, there was a man who had mental issues, he defecates in the bathroom and will go from bed-to-bed harassing patients and all that (IDI 018).*

> *The nurses were okay, they really tried their best the doctors too, were trying because it was really scary that period. They try to make you feel really comfortable. Then there is this woman that usually calls from Irrua she will call you on the phone to know how you are doing and encourage you. The doctors too they were very cooperative but to be sincere, I don't know if it was because of the number of patients things were falling apart, because I told you I stayed very long in the hospital it was not funny . . ..adding patients that were already getting better to patients that were just newly diagnosed and exposing them to re-infection because at one time I started sneezing I was scared that maybe I have picked the something again because of the new persons they brought in to the ward (IDI 021).*

There was a mixed response to feeding in the isolation centre. Food was provided for patients in the isolation centre by the government. It was regular, and some of the interviewees

described it as good. A few respondents reported that patients who had conditions that required a special diet were also attended to accordingly. However, some of the interviewees made provisions for their food through colleagues and relatives for personal reasons. Others said the food was bad, not edible, and useless. This category of respondents reported that no special attention was given to their special diet need.

> *I think the government provided the food, the food aspect the food was ok, and it came regularly (IDI 007)*

> *Interviewer: When you discussed with them that you're diabetic that you can't eat their food, did they change it?*

> *Respondent: of course, Yes, they changed it. When they give people garri, akpu [local food], they'll give me wheat (IDI 004)*

> *My feeding I was very choosy not because their food is not good. . . . I told them that I don't want their food. I told my wife today I want banga [local spicy soup] buy fresh chicken make banga for me she will make it then they will pass it to me I will eat I was just eating and relaxing (IDI 005)*

> *The food was not edible. . . . the food was something I couldn't eat.*

> *Interviewer: you couldn't eat it? So how then did you cope with the food if you couldn't eat it.?*

> *Respondent: Ok! People had to bring food from outside.*

> *Interviewer: They were bringing for you? who are they?*

> *Respondent: My colleagues (IDI 002).*

The duration of stay in the isolation/treatment centre varied between 8 and 30 days.

> *That's what I am saying, I cannot recall but it was up to a month. My result was missing so they had to take another sample, so it caused some delays and all that (IDI 021).*

## Emotional effect/response

The experience of COVID-19 had emotional effects on some of the interviewees. Some were traumatized by discovering that they were positive. Others narrated that they became depressed particularly during the treatment and recovery periods. Fear of not recovering from the illness was a common experience. Others were afraid of long-term complications. A few others experienced neither psychological trauma nor fear of not recovering.

> *It was psychologically draining, that will be my assessment because I had to think about a lot of things, I had some sober reflection. Thinking about the fact that anything could happen just from getting the virus and all that. So, I think it was psychologically draining majorly psychologically draining (IDI 008).*

> *It was the first day I was taken there due to the reaction of my family I was really depressed. When they even checked my BP, the thing was high because of the tension and the rest so, after the 2nd day, the contact tracer that followed them up reassured them that no harm will come before them so and they called me to tell me that everything is ok I became more relaxed (IDI 007)*

*Everything about it was I would just say it was scaring. Then nobody can predict the outcome, and everything was just like you don't know what is going to happen to you, you understand.*

*Interviewer: What was your fear?*

*Respondent: My fear was that I wasn't prepared for death. You know when you are religious and sometimes you get yourself involved in certain things and all of a sudden you are now faced with death and you know you are not prepared, you know. That was my basic fear. I just needed prayers and I really wanted to go back and find God. That was all (IDI 017).*

*I was not really scared because I knew it was not a death sentence. When you follow the trend of the news, you will notice that people were surviving it. Although in that isolation ward, there were cases of death as a result of COVID-19 (IDI 014).*

## Family involvement and response

Many interviewees had their families involved and enjoyed the support of their families all through the time of infection. Some family members did not believe the respondent had the infection, but in some, they feared possible death and stigma. Some of the younger respondents avoided informing their parents that they have been infected until they recovered. One of them said, "I didn't want them to worry" (IDI 009). In one case, the family members were afraid to come near him.

*There were just two around in the country. The other ones outside like my elder brother—he is a Medical Doctor, and he is practicing outside. When I told him, he just advised me. You understand. My sister too wass outside, those ones advised me and told me what I should be doing and pray about it. My younger brother never believed it, he said it is not possible, this is not real, and it is just one of these government scams. My mum took it in great faith and my Dad was there for me all through. He came to visit all the time. He is always there to support me (IDI017)*

*Interviewer: How did your family take the news that you were positive and, in the isolation, centre*

*Respondent: They didn't accept it; in short, they didn't take it well, they were afraid.*

*Interviewer: What were their fears?*

*Respondent: You know the way the media people publicized it, that fear, anyone that carries it they believe that the society will stigmatize you, nobody will want to come close to you because the society sees it as a death sentence. So, anybody that has it, the person is already dead. So, that was their [his family] mindset that the society is going to stigmatize them the neighbours and the rest (IDI007).*

## Experience of stigma and discrimination

Many respondents (n = 12) did not experience any form of stigma from family, colleagues, and friends.

*When I became negative, they asked me to come and stay but I refused I spent an extra one week before meeting them I didn't get any form of discrimination, in fact, they offered to come and be dropping food for me when I was isolating at home, but I refused because I understand the implication (IDI006).*

Others (n = 7) narrated direct and indirect encounters which they perceived as stigmatization and discrimination in forms of avoidance and embarrassing distancing by colleagues, friends, relative's and distant persons, and unwanted publicity.

*. . . well, some people were just keeping distant. They ask me to make a comment and some of the time some told me I was defensive. Asking me to comment could be a stigma in our background . . . somehow, they were trying to stigmatize but I related well. So, when somebody says COVID I will say what is COVID, that thing is malaria and also . . . I may be forced to say that there is some political dimension but that notwithstanding we cannot deny the fact that the ailment is here (IDI005).*

*When I came out [of the isolation centre]. It was more like a joke, but some people were asking: have you tested negative? They want to really know if I have tested negative before they can visit me or before I can stay close to them and I thought it was only normal as they are protecting (IDI008).*

*I won't really know or say but they were protecting themselves, using antiseptic such as izal and disinfecting the house. I learnt that some clothes were burnt, some were soaked in izal and even when I came from the solation centre, people were looking at me from a distance (IDI014).*

*Yes, they showed concern on phone but when I left the isolation center and got home that was when I knew it was real, this stigmatization was real. ehhe now, like usually. Like the day they came to take me they came with a van, so my street people knew that that kind of thing has happened in the house, maybe there is a patient. . . . So, getting back home now, going to buy something, people were avoiding me, even close family members they are not coming close to me to talk that is no longer there, everybody is just keeping a distance . . . only my husband, I can say, my Dad and Mum, even my siblings they were scared but I understand why (IDI021).*

*I was asked to go and when I was going, they asked me to make 1 or 2 comments and I asked them the reason they said it's for their archive, but they violated the research ethics and coincidentally I was home the thing was just flying all over social media it was very embarrassing after some time. I just said well it is unfortunate. I don't have issues if they want to share it, I am quite educated but giving me the impression that is for archive purpose and all that and you went ahead to publish it put it on social media and people are now calling me that they saw this and that . . .*

*Interviewer: From what you are saying now, you experienced some stigma and discrimination how it has been especially with relatives.*

*Respondent: No, it's not a stigma for people that asked me and my wife after they saw the video, we told them it was pneumonia, and it came out the other way and it very embarrassing if you call that stigma, I call it another thing bigger than stigma. . . . my mother is 68 going to 70 she doesn't know the dynamics of the ailment and for her health, we didn't tell her what I was passing through so coming out on the television and now seeing the something that was the problem (IDI005).*

## Financial cost

The cost of care, drugs, tests, and food in the isolation centre was borne by the government.

*The government took care of that we were not asked to pay anything throughout my stay there, I didn't spend 1naira I didn't spend a dime there the food the drugs even the water I was drinking everything was free (IDI007).*

However, many (n = 13) incurred some cost through the purchase of drugs particularly those who were on self-isolation, admitted in the hospitals before being taken to the isolation centre, or had to buy a drug that was not available in the hospital.

*I paid for my drugs that were because if I was admitted to the hospital, the hospital would have covered the cost but because I wanted to be self-isolated, I went to get the drugs to commence my isolation, informed my department, and started isolation (IDI006).*

*I bought a cup of selenium for 5000, a cup of zinc 6000, vitamin a is 650 for one sachet of 10, 2000 for the vitamin E. In fact, altogether, maybe roughly 15,000. For watermelon, apple, oranges, around ten to twelve thousand. I was taking them to augment the tablet so that it doesn't change and become symptomatic (IDI011).*

### Recommendations

The respondents provided suggestions on how to increase awareness about the virus; many respondents were of the view that many Nigerians do not believe the virus exists. Speaking on why people do not believe the virus exists, one of the interviewees attributed it to mistrust of government by Nigerians and the Nigerian culture which permits handling life issues from spiritual perspectives.

*. . . generally, many people don't think coronavirus exist and it really does exist, and I think that the reason for that is mistrust over the years between the citizens and the government and also the Nigerian culture allows some of these things because we see things from the spiritual angle . . . the perception is low. . . (IDI010).*

On protective behaviour, better prevention and control of COVID-19 in Nigeria, some of the interviewees offered recommendations and suggestions drawing from their lived experiences. Their recommendation for prevention and treatment included not taking the virus for granted because the prevalence is still low in Africa. If infected, take vitamins, fruits, and rest, follow the COVID-19 rules recommended by the government and medical personnel and go for testing. Recovered patients should act as agents of awareness and the government should promote more education/awareness through the media, particularly in the local languages. Other recommendations that they proffered included keeping the environment clean, early testing to confirm or rule out COVID-19, and total adherence to the preventive measures.

*I would recommend the person to take vitamins as prescribed then try and rest and take plenty of water and fruits, good fruits to help the immune system (IDI009).*

*Definitely, it has. What changed now is that nonchalant attitude that COVID-19 doesn't exist, and I started telling people that it was real and that they should always take precautionary measures. People should not believe that it was government propaganda, that it was just an opportunity to exploit money (IDI014).*

Also suggested by the study participants was that health workers in the frontline need more training on patient management. Some also suggested personalized care, increase in the

number of doctors and nurses managing patients in the isolation centre, the involvement of private providers in the treatment, increase in the number of testing, private bed space, more and improved facilities at the isolation centres, and supportive interaction from fellow patients and medics. The medical personnel who were interviewees emphasized adherence to the ethics of managing patients, and increased welfare for frontline medical personnel.

*I think there are two dimensions to it first I was marveled at the young nurses working while I was in the teaching hospital unlike me. I was told I had COVID and I know some parameters for me to be stable which I don't think most of those nurses have the information because the fear and what they display was uncalled for. . . .What I think while they are educating the public, they need to think also educate people in the health sector. The management of COVID starts from the management of Health care providers and good credit to them they did their best, but I am convinced if my wife was not there, they said I should change my oxygen mask and the doctor that was there was unwilling to assist. If my wife was not there maybe I would have gone into a coma (IDI005).*

*Yes, like when I was there, they were very professional in reassuring and providing support. I think they need to partition the cubicles because you can see people dying and this made us scared when you ask the providers some questions, they just ignore you and that triggered blood pressure. So probably even with the pressure, they should look for a way `probably because they were understaffed, I don't know because I kept seeing the same face all the time so feel if they were much the stress of the work will not be on them all that (IDI019).*

*They can do better, maybe having separate wards for patients that are already getting well and for the new ones, and they should also work on the food too. There was no light, there was also this light issue; the light was always getting spoiled. At a point, we started protesting. Thenthe toilet was bad, I was even helping to clean the toilet myself because the cleaners whether they were afraid of the COVID or what despite the things they wear they don't come in to clean, so the place is always messed up. So, they can do better (IDI021).*

## Discussion

This study was designed to explore the specific experiences of persons who were infected with COVID-19, but who recovered completely and subsequently tested negative for the virus. A secondary objective of the study was to identify essential elements in the lived experiences of such persons, which would be useful in designing appropriate policies and programs for managing the virus in Nigeria. Although the characteristics of COVID-19 have been published widely in the global literature [20, 21], it is important to investigate how the manifestations of the disease may differ between population groups. In this study, we examined the experiences of the survivors of COVID-19 under six main themes: how they complied with prevention measures, their perceptions on how they contracted the virus, the nature of the symptoms they experienced, the management of the disease and their experiences with the health system, their emotional experiences with friends and relatives, and their recommendations on specific strategies to prevent and manage the virus based on their experiences.

The results showed that although all the respondents knew the recommendations prescribed by the NCDC for the prevention of the virus [22], not all of them fully complied with the recommendations. Indeed, many especially the health professionals indicated that they only partially complied as they broke the protocol when they felt relatively free, especially in clubs, and offices and when in the company of "trusted friends". Furthermore, three

respondents stated that they did not comply with the protocols. This authenticates existing evidence [22, 23] which suggests that poor or inadequate compliance with prevention measures is a leading causative factor for COVID-19. Clearly, more intensive efforts need to be put in place to improve compliance with the prevention protocols recommended by the NCDC.

Regarding perceptions on how the study participants contracted the virus, it was of interest that many could provide vivid narratives. Many of the study respondents reported that they might have contracted the virus from contacts in meetings (especially political meetings), banks, clubs, offices, and while in public transport (taxis). Some specifically reported contracting the virus while on trips outside the country especially in the UK, and Germany, while a few reported that they were not sure how they contracted the virus. This finding further confirms the need for compliance with the NCDC's recommendations of social distancing and sanitization after contacts with friends and relatives.

We used the opportunity of this study to enumerate the symptoms the patients may have had as part of their experience of COVID-19. Although a few of the respondents reported that they had no symptoms whatsoever, several reported diverse symptoms. These include anosmia, general weakness, fever, anaemia, difficulty with breathing, rigor, headache, sore throat, sneezing, "malaria-like symptoms", cold symptoms, and pneumonia. The most commonly reported symptoms were anosmia, fever. and loss of taste, which were reported by nearly all the respondents. These reported symptoms are similar to those that have been reported in other countries [24, 25] and suggest that the Nigerian population of COVID-19 patients may be experiencing the same pattern of illness as elsewhere. Of interest, was that this cohort reported no unusual symptoms. Only three respondents reported pre-existing medical conditions, which included past pneumonia, and diabetes mellitus. Although the study has not investigated a representative sample, it supports the previous reports [26, 27] which suggest that survivors of COVID-19 maybe those without significant pre-existing health conditions.

We solicited information on the nature and characteristics of the clinical management received by the respondents. All tested positive for the virus at health facilities designated by the NCDC for diagnosis. A major feature was that many did not receive formal reports of the tests but were simply informed after days of waiting that they were positive for the virus. This pattern of reporting resulted in mixed responses to the results of the tests including disbelief, confusion, distress, shock, and fear. The delayed receipt of the results also angered some of the respondents as this increased their anxiety level and may have increased their risk of transmitting the virus to others. This is critically important to address in current efforts to prevent COVID-19 transmission in Nigeria. Current diagnostic testing based on polymerase chain reaction (PCR), which tends to be delayed, should be replaced by a model which uses validated rapid diagnostic tests to identify potential positive patients that would then be confirmed later by the more specific predictive test. This model, currently being considered by the NCDC [28], would improve the diagnostic efficiency and efficacy of COVID-19 in the country. Interestingly, the Nigeria COVID-19 Research Consortium (NCRC) is currently undertaking a nationwide validation of COVID-19 antibody and antigen rapid diagnostic kits.

Although a few respondents reported that they self-isolated after the diagnosis of the virus, many were treated in isolation centres that were established across the state by the government. We solicited information on the specific methods of treatment used either when they self-isolated or at the isolation centres. The results showed self-reporting of a wide range of medications including hydroxychloroquine, chloroquine, antibiotics (azithromycin, Augmentin, and erythromycin), vitamins (C, A, D, E, and multivitamins), dexamethasone, selenium, zinc, paracetamol, cough medicines, and anti-diarrheal medications. These are consistent with reports in the literature that have been used globally for the treatment of COVID-19 [29, 30], but it was evident that there were no consistently reported medications that were uniformly

reported by the respondents. When asked to identify what they considered as most effective that led to their recovery, diverse elements were reported by the respondents. These included plenty of sleep and rest, a lot of water, eating well, bathing with warm water, inhalation of steam with herbs, drinking warm water with lime, drinking extracts of bitter leaves, garlic, ginger, vegetables, fruits, and herbal medications (called "agbo"). It was of interest that many reported the use of herbal medications in addition to orthodox medicine, with none reporting the sole use of herbal medicines. In contrast, a few respondents attributed their survival to herbal medicines rather than orthodox treatments.

From the results on treatment methods, it is evident that there is yet no approved protocol for the management of the virus in Nigeria. Although a definite method of treatment has not yet been declared, it is crucially important that a protocol be developed and used for the management of the virus in the country. The protocol should include the various elements identified by the survivors in this study, and from evidence from the international literature. Clinical guidelines from the National Institute of Health in the United States now advice against the use of hydroxychloroquine, chloroquine, Azithromycin, Tocilizumab (Actemra) and other IL-6 inhibitors, and kinase inhibitors) for treating COVID-19 [29, 30]. More effective therapies are now used in high-income countries. For example, Remdesivir (Veklury) is currently the only medication approved in the United States to treat COVID-19, but there are many clinical trials on other potential therapies, such as monoclonal antibodies, convalescent plasma, and immune-based therapy. Scientists are also investigating drugs to prevent COVID-19 before and after exposure to the virus [29, 30].

We investigated the respondents' experiences of the health care system during treatment with COVID-19. This was important to gauge the responsiveness and effectiveness of the health care system to the virus and to identify points requiring remediation. Overall, the health care system was rated by the respondents as good in some domains including the quality and effectiveness of contact tracing, checks made by health professionals, friendliness of health providers, and because treatment was, free. Indeed, many of the respondents posited that their survival was proof of the effectiveness of the health care system. However, several unpleasant experiences were also reported. These included poor transfer/referral mechanisms, providers not always available especially at night, "unlivable" isolation wards, and poor-quality food. It is clearly important that the health facilities managers address these concerns raised by COVID-19 survivors and put in place corrective measures to improve the quality of the clinical management of the disease.

Our study also investigated possible emotional effects the virus exposure may have had on the respondents. This line of inquiry is necessary because previous qualitative studies reported that physicians and nurses who provided care at the early stage of the COVID-19 outbreak in China experienced emotional and psychological stress [31–34]. The psychological effects reported were severe, and timely intervention was advocated [32, 33]. The subjective experiences reported in these studies among healthcare workers at the frontline of the pandemic include negative emotions of fatigue, discomfort, and helplessness caused by high-intensity work, fear and anxiety, and concern for patients and family members. Their coping style includes psychological and life adjustment, team support, and rational cognition.

The patients in our study reported emotional trauma, depression, and fear. Fear was due to consideration for their health and the likelihood that they may not survive the episode. The respondents also reported varying degrees of stigma including avoidance by friends and relatives and unwarranted publicity of their cases. However, these features of stigma did not persist over time and suggest the need to include the provision of emotional and psychological care as part of the support for COVID-19 patients.

The respondents in our study made specific recommendations based on their experiences on ways to improve medical management and reduce the case-fatality from the virus. Their recommendations included widespread information dissemination to reduce the number of persons who do not believe that the virus exists or is real, especially those who interpret the disease as a spiritual problem. In addition, the need to comply with the COVID-19 prevention protocols by all persons and under all settings were strongly recommended by the respondents. They also recommended the training of health personnel to improve the management of the disease, and the involvement of private practitioners in the use of a standard treatment protocol. These recommendations are crucially important and have implications for the design of policies and programs for the prevention and control of COVID-19 in Nigeria.

Most importantly, local research would need to be conducted to monitor the effectiveness of the protocol adopted in Nigeria, and establish the effectiveness of the orthodox and traditional methods of treatment. Indeed, we posit that the effective method of treatment of COVID-19 and other emerging viral illnesses may well lie in traditional (herbal) medicines that are yet to be identified in the country. Efforts should be focused in the coming years to identify effective orthodox and/or traditional methods of treatment of the virus based on empirical evidence.

### Strengths and limitations of the study

To the best of our knowledge, this is the first study to investigate the lived experiences of persons recovering from COVID-19 in Nigeria. Our study participants identified the possible causative pathway, the clinical features, and the methods of management of the virus that led to their recovery. The questions were posed to the study participants in a value-free, open-ended manner, and only persons who agreed to answer the questions truthfully and impartially were interviewed. This greatly increased the internal validity of the study.

The qualitative design of the study limits our ability to make inferences regarding the prevalence of the clinical characteristics and methods of treatment shown. It would have been difficult to get an adequate number of persons for a representative cohort study, and the structured design of the questionnaire would have limited the extent to which the reasons for reported actions are explained. The use of the snowball sampling method in recruiting our study participants limits the external validity of the findings presented. It is impossible to determine the sampling error or deduce inferences about populations based on the obtained sample with snowball sampling. The participants' social networks are usually not randomly drawn. A few cases after referrals, some potential participants refused to participate in the research study. Also, given that people typically refer to those they know and have similar experiences, the snowball sampling strategy is fraught with potential sampling bias and margin of error [19, 35].

Despite the stated limitations in our study, the phenomenological research approach [36] is constructive in offering uniquely rich and impressive perspectives that provide us with a profound and detailed understanding of the lived experiences of persons recovering from COVID-19 as described in this report. Our findings are useful for examining experiences of difficult situations and providing information that could lead to the design of better approaches for addressing the complex multitudes of issues raised in this study.

### Conclusion

The results of this study indicate that persons recovering from COVID-19 in Nigeria had varied experiences relating to the mode of infection, the clinical features, methods of treatment, and psychosocial effects of the virus. These experiences would be useful for the design of

interventions, policies, and programs for restraining the virus in the country. We believe this approach will be valuable and useful in all countries and could improve the equitable and cost-effective use of resources for the management of the disease.

## Supporting information

**S1 File. COREQ checklist.**
(PDF)

**S2 File. Interview guide.**
(DOCX)

## Acknowledgments

We acknowledge the contributions of Dr. Kenneth Maduako, Dr. Francis Igberaese, and Brian Igboin in the data collection.

## Author Contributions

**Conceptualization:** Friday E. Okonofua, Lorretta F. C. Ntoimo, Akhere A. Omonkhua, Joseph Balogun.

**Data curation:** Lorretta F. C. Ntoimo.

**Formal analysis:** Lorretta F. C. Ntoimo.

**Funding acquisition:** Friday E. Okonofua, Vivian I. Onoh, Akhere A. Omonkhua.

**Investigation:** Friday E. Okonofua, Vivian I. Onoh.

**Methodology:** Friday E. Okonofua, Lorretta F. C. Ntoimo, Akhere A. Omonkhua, Christiana A. Alex-Ojei, Joseph Balogun.

**Project administration:** Friday E. Okonofua, Lorretta F. C. Ntoimo, Vivian I. Onoh, Akhere A. Omonkhua.

**Resources:** Friday E. Okonofua, Vivian I. Onoh, Akhere A. Omonkhua.

**Software:** Lorretta F. C. Ntoimo, Christiana A. Alex-Ojei.

**Supervision:** Friday E. Okonofua, Vivian I. Onoh, Akhere A. Omonkhua.

**Validation:** Friday E. Okonofua, Vivian I. Onoh, Akhere A. Omonkhua, Joseph Balogun.

**Writing – original draft:** Friday E. Okonofua, Lorretta F. C. Ntoimo, Vivian I. Onoh.

**Writing – review & editing:** Friday E. Okonofua, Lorretta F. C. Ntoimo, Vivian I. Onoh, Akhere A. Omonkhua, Christiana A. Alex-Ojei, Joseph Balogun.

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
