## [Decision Letter · Decision Letter 0]

28 Jul 2021

PONE-D-21-19514

Lived experiences of recovered COVID-19 persons in Nigeria: A phenomenological study

PLOS ONE

Dear Dr. Okonofua,

Thank you for submitting your manuscript to PLOS ONE. After careful consideration, we feel that it has merit but does not fully meet PLOS ONE’s publication criteria as it currently stands. Therefore, we invite you to submit a revised version of the manuscript that addresses the points raised during the review process.

We look forward to receiving your revised manuscript.

Kind regards,

Jianguo Wang, PhD

Academic Editor

PLOS ONE

2. In the Methods section, please provide additional information regarding the interview guide development process, including the theories or frameworks which were employed. And please include a copy of the interview guide as supporting file.

Reviewers' comments:

Reviewer's Responses to Questions

**Comments to the Author**

1. Is the manuscript technically sound, and do the data support the conclusions?

Reviewer #1: Yes

2. Has the statistical analysis been performed appropriately and rigorously? 

Reviewer #1: Yes

3. Have the authors made all data underlying the findings in their manuscript fully available?

Reviewer #1: Yes

4. Is the manuscript presented in an intelligible fashion and written in standard English?

Reviewer #1: Yes

5. Review Comments to the Author

Reviewer #1: Some figures need to be inserted for example the Ct or MRI of the patients, or x-ray of their chest. Also, some laboratory tests should be presented in a table to be very attractive article with statistical analysis.

6. PLOS authors have the option to publish the peer review history of their article (what does this mean?). If published, this will include your full peer review and any attached files.

Reviewer #1: **Yes: **Heshu Sulaiman Rahman

---

## [Author Response · Author response to Decision Letter 0]

23 Feb 2022

Dear Editor,

Re: PONE-D-21-19514

Lived experiences of recovered COVID-19 persons in Nigeria: A phenomenological study

Thank you for considering our article for publication in your journal. We have now responded to the editorial, and reviewer’s comments. Please, see the point-by-point response below. 

Your Sincerely,

Friday E. Okonofua

Corresponding Author 

Point-by-point response

Response: Our manuscript meets the style requirements.

2. In the Methods section, please provide additional information regarding the interview guide development process, including the theories or frameworks which were employed. And please include a copy of the interview guide as supporting file.

Response: This has been addressed in the methods, and the guide is uploaded as a supporting file. 

Response: Access to the data associated with this article is restricted for ethical reasons due to the potentially identifying and sensitive information about the participants who were COVID-19 patients. However, on request from the corresponding author, some of the data may be shared.

Response: The references have been updated. There are no omitted references 

Reviewers' comments:

Reviewer #1: Some figures need to be inserted for example the Ct or MRI of the patients, or x-ray of their chest. Also, some laboratory tests should be presented in a table to be very attractive article with statistical analysis.

Response: Thank you. However, we are unable to insert the suggested figures because of the following: 1) The study is a qualitative study, and statistical analysis is not appropriate, 2) The Ct or MRI of the patients or their x-rays are confidential documents belonging to the different hospitals, and isolation centre where the respondents were treated. The are not accessible to us.

---

## [Decision Letter · Decision Letter 1]

4 Apr 2022

PONE-D-21-19514R1Lived experiences of recovered COVID-19 persons in Nigeria: A phenomenological studyPLOS ONE

Dear Dr. Okonofua,

Thank you for submitting your manuscript to PLOS ONE. After careful consideration, we feel that it has merit but does not fully meet PLOS ONE’s publication criteria as it currently stands. Therefore, we invite you to submit a revised version of the manuscript that addresses the points raised during the review process.

We look forward to receiving your revised manuscript.

Kind regards,

Jianguo Wang, PhD

Academic Editor

PLOS ONE

Journal Requirements:

Reviewers' comments:

Reviewer's Responses to Questions

**Comments to the Author**

1. If the authors have adequately addressed your comments raised in a previous round of review and you feel that this manuscript is now acceptable for publication, you may indicate that here to bypass the “Comments to the Author” section, enter your conflict of interest statement in the “Confidential to Editor” section, and submit your "Accept" recommendation.

Reviewer #2: All comments have been addressed

2. Is the manuscript technically sound, and do the data support the conclusions?

Reviewer #2: Yes

3. Has the statistical analysis been performed appropriately and rigorously? 

Reviewer #2: N/A

4. Have the authors made all data underlying the findings in their manuscript fully available?

Reviewer #2: No

5. Is the manuscript presented in an intelligible fashion and written in standard English?

Reviewer #2: Yes

6. Review Comments to the Author

Reviewer #2: The authors have addressed and responded to comments raised in the previous review and this has improved upon the quality of the manuscript. However, the authors still need to address a few issues on ethics regarding this study. How did the participants grant a written informed consent when participation is virtual? Also, the authors should include the ethical review number granted by the Ethical Review Committee of the University of Benin.

7. PLOS authors have the option to publish the peer review history of their article (what does this mean?). If published, this will include your full peer review and any attached files.

Reviewer #2: **Yes: **Ismail Ayoade Odetokun

---

## [Author Response · Author response to Decision Letter 1]

18 Apr 2022

Point-by-point response

Journal Requirements:

Response: The reference list is correct and complete. We did not cite any article that has been retracted. 

Reviewers’ comment

Reviewer #2

How did the participants grant a written informed consent when participation is virtual? Also, the authors should include the ethical review number granted by the Ethical Review Committee of the University of Benin.

Response: The consent to participate was obtained from the interviewees at the time of recruitment.

The ethnical approval number has been included in the manuscript. Thank you.

---

## [Decision Letter · Decision Letter 2]

25 Apr 2022

Lived experiences of recovered COVID-19 persons in Nigeria: A phenomenological study

PONE-D-21-19514R2

Dear Dr. Okonofua,

We’re pleased to inform you that your manuscript has been judged scientifically suitable for publication and will be formally accepted for publication once it meets all outstanding technical requirements.

Kind regards,

Jianguo Wang, PhD

Academic Editor

PLOS ONE

Additional Editor Comments (optional):

Reviewers' comments:

Reviewer's Responses to Questions

**Comments to the Author**

1. If the authors have adequately addressed your comments raised in a previous round of review and you feel that this manuscript is now acceptable for publication, you may indicate that here to bypass the “Comments to the Author” section, enter your conflict of interest statement in the “Confidential to Editor” section, and submit your "Accept" recommendation.

Reviewer #2: All comments have been addressed

2. Is the manuscript technically sound, and do the data support the conclusions?

Reviewer #2: Yes

3. Has the statistical analysis been performed appropriately and rigorously? 

Reviewer #2: Yes

4. Have the authors made all data underlying the findings in their manuscript fully available?

Reviewer #2: Yes

5. Is the manuscript presented in an intelligible fashion and written in standard English?

Reviewer #2: Yes

6. Review Comments to the Author

Reviewer #2: All concerns raised have been addressed by the authors. Particularly, issues on ethics have been addressed and the approval number from the institutional review board has been included.

7. PLOS authors have the option to publish the peer review history of their article (what does this mean?). If published, this will include your full peer review and any attached files.

Reviewer #2: **Yes: **Ismail Ayoade Odetokun (Ph.D.)

---

## [Editor Report · Acceptance letter]

3 Aug 2022

PONE-D-21-19514R2 

Lived experiences of recovered COVID-19 persons in Nigeria: A phenomenological study 

Dear Dr. Okonofua:

I'm pleased to inform you that your manuscript has been deemed suitable for publication in PLOS ONE. Congratulations! Your manuscript is now with our production department. 

Kind regards, 

on behalf of

Dr. Jianguo Wang 

Academic Editor

PLOS ONE